# Evaluation of the Perceptions, Attitudes and Practices among Greek Non-Professional Athletes Visiting a Public Hospital during March 2022, towards COVID-19 Vaccination and Its Consequences on Sports Training and Sports Activity

**DOI:** 10.3390/vaccines10111821

**Published:** 2022-10-28

**Authors:** Georgios Marinos, Dimitrios Lamprinos, Panagiotis Georgakopoulos, Nikiforos Kavoukidis, Evangelos Oikonomou, Georgios Zoumpoulis, Gerasimos Siasos, Dimitrios Schizas, Alexandros Nikolopoulos, Petros G. Botonis, Christos Damaskos, Georgios Rachiotis, Pagona Lagiou, Philippos Orfanos

**Affiliations:** 1Department of Hygiene, Epidemiology and Medical Statistics, School of Medicine, National and Kapodistrian University of Athens, 5 Mikras Asias Street, 11527 Athens, Greece; 2Emergency Care Department, Laikon General Hospital, 11527 Athens, Greece; 3Department of Cardiology, Sismanoglio General Hospital, 15126 Amarousion, Greece; 4First Department of Cardiology, Hippokration General Hospital, Medical School, National and Kapodistrian University of Athens, 11527 Athens, Greece; 5Third Department of Cardiology, Thoracic Diseases General Hospital Sotiria, Medical School, National and Kapodistrian University of Athens, 11527 Athens, Greece; 6First Department of Surgery, National and Kapodistrian Laiko General Hospital, University of Athens, 11527 Athens, Greece; 7School of Physical Education and Sport Science, National and Kapodistrian University of Athens, 11527 Athens, Greece; 8Department of Sports Medicine and Biology of Exercise, Faculty of Physical Education and Sport Science, National and Kapodistrian University of Athens, 11527 Athens, Greece; 9Renal Transplantation Unit, Laiko General Hospital, 11527 Athens, Greece; 10N.S. Christeas Laboratory of Experimental Surgery and Surgical Research, Medical School, National and Kapodistrian University of Athens, 11527 Athens, Greece; 11Department of Hygiene and Epidemiology, Faculty of Medicine, University of Thessaly, 41500 Larissa, Greece

**Keywords:** COVID-19, vaccination, athletes, sports activity, attitudes, perceptions

## Abstract

Sports have been majorly impacted by the COVID-19 pandemic. After the lockdown period, vaccination and protocols were implemented to return to normality. We aimed to assess the attitudes and practices related to COVID-19 vaccination among athletes, and to record adverse effects of vaccination, if any. A questionnaire was distributed to 1012 male and female athletes, 15+ years old, within the region of Athens. Vaccination coverage with at least one dose was 93.5%, whereas 53.9% were fully vaccinated. More than half of the participants were infected with SARS-CoV-2 at the time of the study. More than 90% of the participants, considered the vaccines as safe, effective and important for public health. Concern about potential side-effects was raised especially by women athletes (59.1% of women compared to 42.2% of men, *p* < 0.001). The main reasons for avoiding vaccination were fear of vaccine safety, concern about the short time period for vaccine development and testing and doubt of risk of being exposed to SARS-CoV-2 infection. The main reported side-effects were pain at the injection site, fatigue, fever and headache. Approximately two thirds of the participants reported that vaccination did not affect their training, and none reported missing participation in scheduled athletic events. Participants reported high compliance to preventive measures by themselves and fellow athletes, but low satisfaction regarding the implementation of public protocols and the flow of information provided by the authorities. Athletes of older age and those less concerned about potential side-effects were more likely to get fully vaccinated. Nevertheless, the vast majority of the athletes in our study were vaccinated for COVID-19 despite any hesitation regarding effectiveness, safety, or potential side-effects from the vaccines.

## 1. Introduction

The 21st century brought amongst others a great new challenge to the world: SARS-CoV-2 virus, responsible for COVID-19 disease. The worldwide management of the COVID-19 pandemic was focused on the implementation of prevention protocols, physical distancing measures and strict lockdowns that were supported with the enhancement of vaccination. The vast majority adopted community isolation measures to increase social distancing, such as mandatory lockdowns, isolation periods, and closing of public areas such as sports centers and gyms. Preoccupation with COVID-19 and negative changes in lifestyle due to COVID-19 were associated with depression, anxiety and stress in the population [1,2]. Governments in different countries implemented a variety of plans in order to limit the strict lockdowns that included policies and guidance for the re-opening of public places for outdoor activities and sports, including athletics and swimming-related activities. Development and efficacy of several vaccines, and their proven safety constitute significant factors for controlling the pandemic [3,4].

In sports, the COVID-19 pandemic has limited activities considerably and led to the postponement and cancellation of a vast number of national and international events once transmission among athletes was obvious. Similar to the general population, athletes had to adjust to these strict measures by changing their lifestyle, their training routine and deal with the mental and psychosocial impact of the strict rules [5]. The level of the athletes’ adjustment to these measures as well as their practices towards vaccination and isolation instructions were essential not only for the retention of their physical activity but for the limitation of SARS-CoV-2 spread as well. Moreover, the fact that well known athletes from different sports were found to be positive for the disease COVID-19, despite their recognized high fitness levels indicated that athletes were vulnerable to the virus. Therefore, increased fear led to a greater restriction of physical practice for numerous athletes [6,7].

With billions of doses having been administered worldwide, COVID-19 vaccines have been proven to be safe and effective; Sports Medicine Societies supported vaccination in athletes without contraindications. Even though the vaccination has been proven to be a safe and effective practice against severe illness, there was a noticeable non intention from individuals to get vaccinated due to concerns about vaccine side-effects and safety, effectiveness, as well as a lack of trust in the creation process [1,8]. Some medical conditions, such as intestinal diseases, which are highly associated with colorectal cancer [9], are also found to be negative factors for accepting COVID-19 vaccination, despite the recommendations of the experts [10,11,12]. According to Narduci et al. the COVID-19 vaccine hesitancy seems to be more common among individuals younger than 60 years, who have lower levels of education, lower household income, rural residence, and lack of health insurance. The way to effectively manage and retain sports training in the pandemic has been a challenge not only for the professional athletes but for non-professionals as well [13]. The aim of this study was to assess the perceptions, the attitudes and practices related to COVID-19 vaccination among non-professional athletes as well as to record adverse effects of vaccination.

## 2. Materials and Methods

The study was conducted in a large sample obtained from patients visiting LAIKO general hospital, one of the biggest hospitals in the region with a specialized clinic to musculoskeletal injuries, to be examined, from 20th of February 2022 to 20th of March 2022. The patients who were doing sports were asked and accepted to participate in our study voluntarily. Responses were carefully reviewed to ensure that responses were not repeated. The members of the research team who were assigned to distribute the questionnaire to the participants were associated in any way with the responders. The study was approved by the Hospital Ethics Committee and was conducted according to the principles of the Declaration of Helsinki of 1975, as revised in 2008. Common exclusion criteria include the presence of a severe and uncorrectable cognitive, visual, or hearing impairment that would preclude a participant’s ability to complete the questionnaire and lack of exercise. Participants were not required to record their personal details. During the one-month period, a total of nearly 5000 patients visited the hospital and among them it is estimated that approximately ¼ were actively involved with sports. A final sample of 1012 athletes participated to the study. The questionnaire included a minimum set of demographic parameters along with 15 questions regarding practices, attitudes and adverse events related to vaccination and was estimated to take less than 10 min to complete.

The questions were based on previous questionnaires [14,15,16,17]. The questionnaire was forward translated from English to Greek by three independent translators. Greek was the mother tongue of these translators. The prefinal version of the translated questionnaire was tested on a small group of 10 people from the target population. After completing the questionnaire, the respondents were interviewed to ensure that the questionnaire was understandable and presented accurately the meaning of the original questions. The questionnaire was finalized after the cognitive debriefing results were reviewed and the tool was proofread. Concerning the backwards translation of the study instrument, which was modified and included additional questions, no communication was made between the research team and the original authors of the main set of questions for review purposes.

The questionnaire included questions on demographics (sex, age, sport) and perceptions of the importance of vaccination, safety and effectiveness of vaccines. In addition, the questionnaire included questions regarding COVID-19 vaccination, COVID-19 infection and influenza vaccination coverage for flu season 2021-22 (“Have you been vaccinated against COVID-19?” Answer options: Yes/No; “Have you been infected by SARS-CoV-2” Answer options: Yes/No; “Have you been vaccinated with the influenza vaccine for season 2021-22” Answer options: Yes/No). Vaccination coverage against COVID-19 included the receipt of one, two or three doses of a vaccine. In the case of COVID-19 vaccination refusal, the participants were asked to report the reason of non-vaccination (I have concerns for the vaccine safety; I am not in danger of COVID-19 disease; The time of vaccine development was short; I am using homeopathy drugs; Pending appointment; I am opposite to vaccination). Participants were asked to rate on a four-point Likert scale (answer options: Fully agree, agree, disagree, and fully disagree) possible concerns over COVID-19 vaccination side effects, the importance, effectiveness, and safety of vaccinations. Moreover, the respondents were asked to evaluate the quality of COVID-19 prevention protocols information and implementation from the public state and satisfaction from the implementation of those protocols from coaches, fellow athletes and themselves. The questionnaire included also questions about possible side effects from vaccination and athletes were asked to state if these affected their training leading to loss of the training or an official event. Lastly, the participants were asked about their sources of information on the COVID-19 vaccines.

For descriptive purposes the participant characteristics were presented as absolute (N) and relative frequencies (percentages; %) except for age which was presented as mean and standard deviation. The chi square test was used to determine statistically significant differences among subgroups of the study’s participants and the *t*-test for age only. A chi square test of linear trend was applied to examine trends in proportions of reported side-effects across the 3 doses or of the satisfaction levels. Multiple logistic regression models were applied separately for males and females to estimate the odds of getting fully vaccinated (3 doses compared to none, one or two doses) by age, type of sports, concerns about side-effects and source of information on vaccination. Sex-specific odds ratios (OR) and 95% confidence intervals (95%CI) are reported to depict the magnitude of the associations. Differences were considered as statistically significant at a *p*-value < 0.05. SPSS 23 for Windows was used for the analyses.

## 3. Results

The majority of the study sample were vaccinated for COVID-19 (N = 946, 93.5%) and more than half had taken three doses (N = 550, 54.4%) (Figure 1). 

Participant characteristics and their crude association with vaccination coverage (whether vaccinated or not) are presented in Table 1. The mean age of the 1012 athletes was 24.4 years (sd = 8.0), and female athletes were much younger than males (27.6 ± 9.6, range: 15 to 53 years for males and 20.8 ± 2.7, range: 15 to 29 years for females). Four out of 10 were exercising with swimming (42.4%) and other settings reported included gymnastics, team sports, combat sports and other (including skiing, climbing, dancing, running, hammer throw). A small percentage of the athletes (N = 143, 14.1%) were vaccinated for influenza. More than half of the participants (N = 583, 57.6%) were infected from SARS-CoV-2 at the time of the study. Univariate analysis showed that mainly males of higher age, not infected from COVID-19 and vaccinated for influenza were significantly associated with COVID-19 vaccination.

Based on their statements the main source of information regarding COVID-19 vaccination was from an official public health source (International Health Organizations (WHO, CDC, eCDC) (N = 242; 23.9%), Greek National Public Health Organization (N = 121; 12%), National Medical Association (N = 33; 3.3%), scientific sources/publications (N = 132; 13%), media (N = 242; 23.9%), social media (N = 165; 16.3%) and other sources (N = 77; 7.6%). Overall, 52.2% were informed from official scientific sources while 47.8% from social media and TV.

The participants that were not vaccinated (N = 66, 6.5%) stated that this was due to fears for the vaccination safety (N = 11, 16.6%), expressed concerns for the short time period for vaccine development and testing (N = 11, 16.6%) and doubted for the potential risk of being infected (N = 22, 33.4%) (Table 2). 

Athletes who had been vaccinated appeared to believe that vaccines were essential for public health and safe and seemed to be less worried about potential side-effects. In accordance, the percentage of vaccinated athletes that characterized vaccination as an effective method for public health protection was slightly higher but not statistically significant, compared to those who did not agree with that statement (data not shown). The participants’ beliefs towards vaccines are presented in Table 3, by sex. Female athletes tended to believe that vaccines are effective (*p* = 0.005) but on the other side they were more worried about side-effects than men (59.1 vs. 42.2%, respectively, *p* < 0.001).

Information regarding the satisfaction of the participants for the implementation of prevention protocols in training centers is presented in Table 4. Most of the athletes stated higher levels of satisfaction as it regards to their personal or their fellow athletes’ and coaches’ compliance to protocols (*p* for trend <0.001). On the contrary, they reported bad or neutral feelings regarding the implementation of public protocols and the flow of information provided by the authorities. Female athletes were more satisfied than men concerning their personal compliance (84 vs. 73%, respectively, *p* < 0.001) while men considered their fellows’ compliance at a higher level of satisfaction compared to women (46 vs. 36%, respectively, *p* = 0.003) (data not shown). 

Table 5 presents the main side-effects in relevance to the sequence of the vaccination dose as reported by the athletes. The main reported side-effects were pain in the injection site, fatigue, fever and headache. However, a noticeable decrease of reported side-effects was noticed from 1st to 3rd dose for most of them (*p* for trend <0.05) except for myalgia and sickness which were reported from approximately 1 out of 10 individuals, irrespective of dose sequence. Nevertheless, and irrespective of sex, 36% of the participants (N = 341 out of 946) reported that the vaccination affected their training leading to loss of the training, while none of the athletes reported missing participation in official athletic events due to vaccination side-effects (data not shown).

Table 6 presents the odds ratios of getting fully vaccinated, versus not getting fully vaccinated (3 doses vs. none or max of two), by the listed factors, separately for male and female athletes. Among males, younger athletes, involved in swimming or combat sports were less likely to get fully vaccinated. Moreover, those with more concerns about potential side-effects from vaccination had lower odds by approximately 39% to get fully vaccinated compared to those with less concerns (OR = 0.61; 95%CI: 0.39 to 0.93). The same results were more striking among female athletes, with the exception of female athletes engaged in combat sports, who were more likely to get fully vaccinated, although the result did not reach statistical significance (OR = 1.83; 95%CI: 0.62 to 5.38). Additionally, female athletes involved in “other” sports (like skiing or dancing, etc.) were approximately 3 times more likely to get all three doses compared to those in gymnastics or the remaining types of sports (OR = 2.79; 95%CI: 1.30 to 5.98). Information obtained from social media, compared to information from scientific sources, tended to be associated with lower likelihood of athletes getting fully vaccinated but the association did not reach statistical significance. 

## 4. Discussion

The vast majority of the athletes in our study (93.5%) were vaccinated for COVID-19 despite any hesitation regarding effectiveness, safety, or potential side-effects from the vaccines. Athletes of younger age, being involved in swimming and combat sports (only men) who are highly concerned about potential side-effects from vaccination and are informed from sources other than scientific, were less likely to get fully vaccinated. Most of the athletes considered the vaccines as safe, effective, and important for public health but a concern about potential side-effects was raised especially by female athletes. The main reported side-effects were pain at the injection site, fatigue, fever and headache. Two out of three athletes stated that vaccination did not affect their training, and no one reported missing of any official game. The participants reported high compliance to preventive measures by themselves as well as their fellow athletes and coaches, but low satisfaction as it regards to the implementation of public protocols, or the official information provided by the authorities.

Vaccination reluctancy rates especially among the younger population groups vary greatly, depending on a number of reasons with psychological variables and communication frames to stand as key determinants. In a recent Italian study, the authors showed that depending on the goal (stimulate vaccination intention, vaccination trust, or vaccine attitude), not all the communication stimuli are equally effective on this target population [18]. In line with our study findings, trust and attitudes towards vaccines, concern about the pandemic have been found to be effective predictive variables of people’s intention towards vaccination. In a Greek study during the period just after the development of COVID-19 vaccines, a significant proportion of individuals in the general population were unwilling to receive a COVID-19 vaccine [19]. According to our findings, this trend has changed, and the majority of the athletes were vaccinated.

We found that despite the high vaccination rate, half of the athletes reported fear for possible side-effects from COVID-19 vaccination. Tanaka et al. supported that messages that emphasize the majority’s intention to vaccinate and scientific evidence for the safety of the vaccination had the strongest positive effect on the willingness of young people to be vaccinated [20]. This is also supported from evidence that the individual willingness to get vaccinated is correlated with socio-demographic and psychological characteristics of the respondents [21]. We found that information obtained from social media might discourage individuals to get fully vaccinated. Recent studies have shown that online misinformation and especially misinformation regarding the impact of COVID-19 vaccine on future fertility is creating false fears in women towards the potential occurrence of side-effects of the vaccine [22,23].

The type of sport activity seemed to play a role in the likelihood of getting fully vaccinated. Combat sports seemed to be a negative factor for vaccination acceptance among male athletes. The high levels of self-confidence, the mental toughness and the perfectionism that those athletes need to have to be successful and the feeling that no one can defeat them could be some of the reasons for not taking the vaccine [24]. Similar results were recorded for swimming. The information that water-borne transmission is still not certain, the reduction of the likelihood to get infected due to the use of chlorination of swimming pool and all the protective strategies in order to perform swimming activities together with the concerns about potential side-effects could dissuade the participants to get vaccinated [25,26]. McGuine et al. noted that team sport athletes reported lower levels of physical activity, and worse Health-related quality of life (HRQoL) than athletes who were involved in individual sports or in both individual and team sports [27]. In Greece, individual sport athletes were able to continue training in their sports when physical-distancing restrictions were implemented and, thus, were not affected to the same extent as team sport athletes. Therefore, team sport athletes probably experienced greater limitations that led them to reduced physical activity. A tendency towards the same direction was found in our study although it did not reach statistical significance. We further found that participants, especially women, competing in “other” sports including skiing or dancing, were more likely to get full vaccinated. Ski resorts and dancing schools are considered overcrowded places where face-to-face activities take place and COVID-19 could be transmitted more easily [28]. 

The athletes were asked to state whether they experienced any side-effect related to their SARS-CoV-2 vaccination. In our study, we found a low rate of mild side-effects. This is in accordance with previous studies in athletes that also found generally mild effects do not typically impact routine activities and are more prevalent in younger athletes [29,30]. The incidence of side-effects in our sample was lower and a noticeable decrease of reported side-effects was noticed from 1st to 3rd dose. Only one third of the participants stated that they had to stop training for a period of time due to the vaccination side-effects. Nevertheless, none of them stated the loss of official games due to vaccination side-effects. 

Among athletes, their desire to retain their training status seems to be an important determinant of their positive attitude towards vaccination and compliance to COVID-19 prevention protocols and this may explain the higher rates of vaccination in comparison to previous studies indicating a rate of 70–75% in the general population worldwide [31]. Moreover, a small percentage of the athletes were vaccinated for influenza, had been also vaccinated for SARS-CoV-2. This finding is in accordance with recent studies where there is clear evidence that the COVID-19 pandemic had a positive impact on the acceptance of influenza vaccination by the general population in Greece [32,33,34]. The rate of influenza vaccination in our study among athletes was quite low and that may be explained by the fact that the majority of the participants are healthy adults that are not included in the “high risk” groups. One other possible explanation is that influenza vaccination rates prior to the pandemic in Greece among healthy adults were low [35]. 

Several studies noted that the level of professional athletes’ satisfaction seems to be interrelated with the effectiveness of the protocols not only in terms of safety and non-infection but also in terms of training opportunities. The compliance to the strict COVID-19 prevention protocols had a clear impact on the training schedules of elite athletes. As a consequence, this seems that it has affected their sleeping habits, caused unhealthy habits and coping mechanisms such as increasing their carbohydrate intake and preferring sedentary behavior above active behavior [7,36]. Similarly, COVID-19 lockdown and strict prevention protocols had a major impact to the general population, affecting eating patterns, body weight, mental health and physical activity [37,38]. Neutral feelings occurred concerning the implementation of prevention protocols in training centers during the COVID-19 pandemic. The success of a government’s control strategy relies on public trust and broad acceptance of response measures. Athletes need to adhere to government restrictions when training, but they should be actively supported in continuing to exercise. Therefore, a safe sporting environment for athletes and coaches is needed. Until the effective management of the pandemic, limitations on sporting participation will continue [39]. 

Moreover, we found that the satisfaction from the information regarding these protocols from the public state were also low. It is generally acceptable that the success of public interventions and policies designed to reduce the impact of the COVID-19 pandemic depends on how well individuals are informed about both the consequences of infection and the steps that should be taken to reduce the impact of the disease [40]. Based on their statements the main source of information regarding COVID-19 vaccination in our study was from an official public health source and scientific publications, whereas near half of them were informed through media, mass media and other sources. 

The study’s participants stated increased satisfaction from their fellow athletes and/or coaches’ compliance to preventive measures and even greater satisfaction from their personal compliance to measures, which were mostly women. The low rates of sports-related COVID-19 transmission indicate in a way the compliance of athletes with the prevention measures. In line with our findings regarding satisfaction, Vitali et al., in their study analyzing the effects of the COVID-19 pandemic on well-being and sport readiness, concluded that female athletes showed higher perceived COVID-19 risk [41].

As we move on in the era of COVID-19, the return to normalcy is signified by the restart of major athletic events, like the Olympics, allowing the elite athletes to reach again the training levels before the pandemic and continue their careers. Similarly, post-lockdown physical activity is also increased in the general population by slowly returning to the pre-pandemic lifestyle and physical activity level [37].

This study has a number of limitations. Firstly, the participants consisted mainly of amateur athletes, hence the results cannot necessarily be extrapolated to professional athletes, for whom pandemic restrictions may have different impact as indicated in recent literature [36,42,43]. Secondly, the cross-sectional nature of the study could not enable us to differentiate cause and effect. Thirdly, our study was questionnaire-based, thus information bias, as well as recall bias cannot be excluded. Furthermore, we were not able to obtain responses from non-respondents and this may be a source of selection bias. Lastly, our questionnaire included a small number of questions, and this may impact on internal consistency [44]. Of note, volunteers who participated in the study may be more health conscious, and therefore more prone to getting vaccinated. Finally, for both COVID-19 vaccination and influenza vaccination during the study period, we captured views in a specific time period without taking into account the dynamic nature of the pandemic. 

## 5. Conclusions

In conclusion, the amateur athletes in our study showed a high vaccination rate despite their hesitance regarding effectiveness, safety, or potential side-effects from the vaccines. Age, type of sport activity, concerns of potential side-effects and source of information were important predictors of getting fully vaccinated. Widespread vaccination against COVID-19 is necessary to reduce the disease burden of the SARS-CoV-2 pandemic and for the return to normal sports activities. The low incidence of vaccination side-effects and the narrow impact on training loss seems to stand as a supporter to their attitude and enhanced their satisfaction from their compliance as well as from their fellow athletes and coaches regarding prevention measures during physical activity. The strict restrictions that were engaged from the government in Greece and the loss of training during prolonged periods, as well as cancellation of sport tournaments in many sports, seems that have affected their satisfaction regarding the public measures’ performance and information. This study, despite its limitations, gives an insight regarding Greek athletes’ beliefs attitudes and practices towards COVID-19 vaccination and its consequences on sports training and activity.

## Figures and Tables

**Figure 1 vaccines-10-01821-f001:**
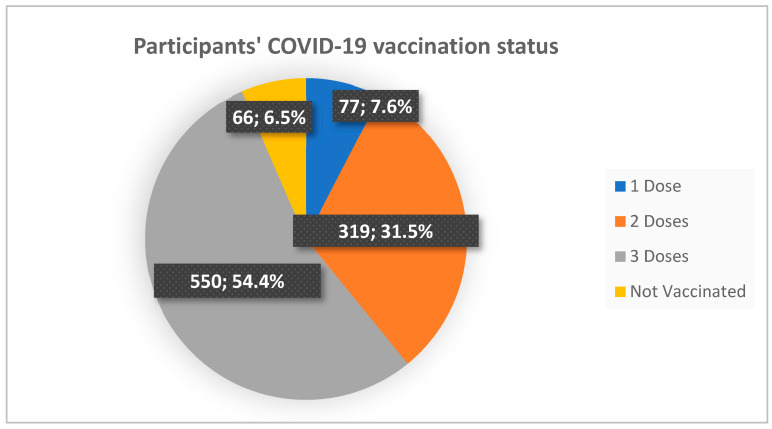
COVID-19 vaccination status of the 1012 study participants.

**Table 1 vaccines-10-01821-t001:** Characteristics of 1012 participants by COVID-19 vaccination status *.

Characteristics		COVID-19 Vaccination Status *				*p*-Value **
	Total (N = 1012)	Yes (N = 946)		No (N = 66)		
	M (±SD)	Mean	SD	Mean	SD	
**Age in years**	24.4 (±8.0)	24.6	8.2	22.3	3.2	0.027
	N (%)	N	%	N	%	
**Sex**						<0.001
Male	545 (53.9)	523	96.0	22	4.0	
Female	467 (46.1)	423	90.6	44	9.4	
**Sports**						<0.001
Gymnastics	264 (26.1)	253	95.8	11	4.2	
Swimming	429 (42.4)	407	94.9	22	5.1	
Combat sports	66 (6.5)	44	66.7	22	33.3	
Team sports	132 (13.0)	132	100.0	0	0.0	
Other (i.e., skiing, climbing, dancing, running, hammer throw)	121 (12.0)	110	90.9	11	9.1	
**Infected by SARS-CoV-2**						<0.001
Yes	583 (57.6)	517	88.7	66	11.3	
No	429 (42.4)	429	100.0	0	0.0	
**Vaccinated for Influenza**						<0.001
Yes	143 (14.1)	143	100.0	0	0.0	
No	869 (85.9)	803	92.4	66	7.6	

* Vaccinated at least with 1 dose. ** *p*-value from chi square test (or *t*-test in the case of age only).

**Table 2 vaccines-10-01821-t002:** Stated reasons for not getting vaccinated for COVID-19.

Reasons	N	%
I am not in danger of COVID-19	22	33.4
Vaccination safety concerns	11	16.6
The time of the development of the vaccines was short	11	16.6
No answer	22	33.4
Total	66	100.0

**Table 3 vaccines-10-01821-t003:** Athletes’ beliefs on vaccines for SARS-CoV-2 by sex.

	Sex		
Beliefs	Males (%)	Females (%)	*p*-Value *
	N = 545	N = 467	
The vaccines are important for Public Health Fully Agree/Agree Fully disagree/Disagree	545 (100.0) 0 (0.0)	456 (97.6) 11(2.4)	<0.001
In general, vaccines are safe Fully Agree/Agree Fully disagree/Disagree	512 (93.9) 33 (6.1)	445 (95.3) 22 (4.7)	0.347
In general, vaccines are effective Fully Agree/Agree Fully disagree/Disagree	472 (86.6) 73 (13.4)	430 (92.1) 37 (7.9)	0.005
I am concerned about vaccination side-effects Fully Agree/Agree Fully disagree/Disagree	230 (42.2) 315 (57.8)	276 (59.1) 191 (40.9)	<0.001

* *p*-value obtained from chi square test.

**Table 4 vaccines-10-01821-t004:** Level of athletes’ satisfaction regarding different aspects of prevention protocols implemented in training centers.

Questions	Dissatisfied (%)	Neutral (%)	Satisfied (%)	*p* for Trend *
How satisfied are you regarding the implementation of prevention protocols in training centers during the COVID-19 pandemic?	231 (22.8)	528 (52.2)	253 (25.0)	0.300
How satisfied are you regarding the information on these protocols provided by the authorities?	363 (35.9)	407 (40.2)	242 (23.9)	<0.001
How satisfied are you regarding the compliance of your fellow athletes and/or your coaches to these protocols?	187 (18.5)	407 (40.2)	418 (41.3)	<0.001
How satisfied are you regarding your personal compliance to these protocols?	66 (6.5)	154 (15.0)	792 (78.3)	<0.001

* *p*-value obtained from the Chi-square test of linear trend.

**Table 5 vaccines-10-01821-t005:** Main side-effects of SARS-CoV-2 vaccination reported by the athletes.

Side-Effect	1st Dose (%) (N = 946)	2nd Dose (%) (N = 869)	3rd Dose (%) (N = 550)	*p* for Trend *
Pain or edema in the injection site	209 (22.1)	231 (26.6)	88 (16.0)	0.039
Fatigue	176 (18.6)	187 (21.5)	77 (14.0)	0.085
Headache	143 (15.1)	110 (12.7)	55 (10.0)	0.004
Myalgia	110 (11.6)	99 (11.4)	55 (10.0)	0.365
Shivering	99 (10.5)	66 (7.6)	33 (6.0)	0.002
Fever	143 (15.1)	165 (19.0)	44 (8.0)	0.003
Arthralgia	110 (11.6)	77 (8.9)	22 (4.0)	<0.001
Redness in the injection site	55 (5.8)	44 (5.1)	0 (0.0)	<0.001
Nausea	55 (5.8)	11 (1.3)	0 (0.0)	<0.001
Lymph nodes swelling	22 (2.3)	44 (5.1)	22 (4.0)	0.040
Sickness	110 (11.6)	77 (8.9)	55 (10.0)	0.206
Pain in the limbs	44 (4.7)	44 (5.1)	11 (2.0)	0.030
Insomnia	11 (1.2)	11 (1.3)	0 (0.0)	0.044
Itching	22 (2.3)	0 (0.0)	0 (0.0)	<0.001

* *p*-value obtained from the chi square test of linear trend.

**Table 6 vaccines-10-01821-t006:** Sex-specific odds ratio (OR) and 95% confidence intervals (CI) for getting fully vaccinated versus not getting fully vaccinated.

	Males (N = 545)			Females (N = 467)		
Variables	OR	95%CI	*p*-Value	OR	95%CI	*p*-Value
**Age (per year)**	1.08	1.05 to 1.11	<0.001	1.23	1.13 to 1.35	<0.001
**Sports**						
Gymnastics	1			1		
Swimming	0.55	0.33 to 0.94	0.028	0.33	0.18 to 0.60	<0.001
Combat sports	0.02	0.01 to 0.16	0.001	1.83	0.62 to 5.38	0.274
Team sports	0.62	0.30 to 1.25	0.179	0.75	0.37 to 1.54	0.437
Other	1.07	0.51 to 2.24	0.868	2.79	1.30 to 5.98	0.008
**Concerns of side-effects**						
No	1			1		
Yes	0.61	0.39 to 0.93	0.023	0.17	0.10 to 0.29	<0.001
**Source of information**						
Scientific sources	1			1		
Social media	0.72	0.48 to 1.08	0.115	0.69	0.44 to 1.08	0.102

## Data Availability

The study data are available from the corresponding author upon reasonable request.

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
