# Peer review of "Evaluation of the Perceptions, Attitudes and Practices among Greek Non-Professional Athletes Visiting a Public Hospital during March 2022, towards COVID-19 Vaccination and Its Consequences on Sports Training and Sports Activity"

_vaccines, 2022, doi:10.3390/vaccines10111821_

Round 1
Reviewer 1 Report
I would like to recommend this meaningful study for publication after minor revision:
1. The format of P-value and p value should be uniformed.
2. The reference styles are confusing and not the standard format of MDPI. Please refer to the template for modification.
3. The Conclusion is too short, please further enrich the content.
4. Some diseases, such as intestinal diseases < Xiaojing Sun, Zhonghua Xue, Aqeela Yasin, et al., Colorectal Cancer and Adjacent Normal Mucosa Differ in Apoptotic and Inflammatory Protein Expression, Engineered Regeneration 2 (2022) 279-287.> , may also affect people's attitude towards the new crown vaccine. Even though it is less likely that this will happen to athletes, it is recommended to briefly describe it in the Introduction.
5. In Figure 1, what’s the difference between the three groups of Orange, Blue and Grey colors.
Author Response
Thank you for giving us the opportunity to submit a revised draft of our manuscript titled “Perceptions, attitudes, and practices of Greek Athletes towards COVID-19 vaccination and its consequences on sports training and sports activity”/ with proposed new title “ Evaluation of the perceptions, attitudes, and practices among Greek non-professional athletes visiting a public hospital during March 2022, towards COVID-19 vaccination and its consequences on sports training and sports activity”, to Vaccines Special Issue "Epidemiology, Vaccination and Public Health". We appreciate the time and effort that you and the reviewers have dedicated to providing your valuable feedback on our manuscript.
. Point 1: The format of P-value and p value should be uniformed.
Response 1: Thank you for pointing this out. We agree with this comment. Therefore, we have modified the format throughout the manuscript.
Point 2: The reference styles are confusing and not the standard format of MDPI. Please refer to the template for modification.
Response 2: We agree with your comment and have incorporated your suggestion throughout the manuscript. REVISED (REFERENCES, p.12-15, lines 429-560)
Point 3: The Conclusion is too short, please further enrich the content.
Response 3: As suggested by the reviewer, we have accordingly modified the conclusions paragraph. REVISED (CONCLUSIONS, p.11, lines 391-405)
Point 4: Some diseases, such as intestinal diseases < Xiaojing Sun, Zhonghua Xue, Aqeela Yasin, et al., Colorectal Cancer and Adjacent Normal Mucosa Differ in Apoptotic and Inflammatory Protein Expression, Engineered Regeneration 2 (2022) 279-287.> , may also affect people's attitude towards the new crown vaccine. Even though it is less likely that this will happen to athletes, it is recommended to briefly describe it in the Introduction.
Response 4: We thank the Reviewer for this comment. To best accommodate her/his proposal, we have accordingly modified the introduction paragraph. REVISED (INTRODUCTION, p.2, lines 91-94)
Point 5: In Figure 1, what’s the difference between the three groups of orange, blue and grey colors.
Response 5: The three groups are referring to the vaccination coverage (number of doses) of the participants that we recorded at the time of the study. Blue color corresponds to 1 dose of the vaccine, Orange corresponds to 2 doses of the vaccine, Grey corresponds to 3 doses of the vaccine. To accommodate your comment as clearly as possible, we have modified the aforementioned Figure by adding an explanation on the right side. REVISED (RESULTS, p.4, line 169-170)
We look forward to hearing from you in due time regarding our submission and to respond to any further questions and comments you may have. We thank the reviewer for his/her thoughtful and thorough review and believe his/her input has been invaluable to make our manuscript more balanced.
Sincerely,
The corresponding author on behalf of all authors
Reviewer 2 Report
First of all, I am grateful for the opportunity to review this paper. Physical inactivity is highly spread worldwide, and a significant part of the global population does not achieve the recommended amount of weekly PA for health. In particular, the pandemic and the related control measures had great consequence on their activivities. In this context, aim of the paper under review is to assess the attitudes and practices related to COVID-19 vaccination among athletes, and to record adverse effects of vaccination. In Greece.
The subject under study is certainly important, especially in the historical period we are experiencing. The article presents interesting results but, it must be improved, especially for some methodological concerns.
Title: It should be improved, highlighting the main object of the study, period and population.
Introduction: The authors should make it clear about what is the gap in the literature that is filled with this study. The study must be actualized by framing it within the vast body of literature that addressed the COVID-19 related changed habits in the context of the main 21st-century challenges (refer to articles with DOI: 10.3390/ijerph191911929).
Methods: This section raises the main scientific concerns. The questionnaire is a non-standard tools. The use of an unreliable instrument is a serious and irreversible limitation. The fact that a similar methods have been used in previous studies is not sufficient. A validation process must be performed and reported to evaluate the tool. What about intelligibility, reliability and validation index?
The authors do not propose a minimum sample size, what is the reference population? The Authors talk about patients of an hospital, what is the study population?
Without the numerical identification of the reference population is not clear the validity of the study. A non-representative sample is by its self a non-sense-survey. How did they avoid selection bias? How was questionnaire distributed? It is not clear how participants were recruited? Did thy Authors pay some company to provide lists of subscribers or distribute the investigation tool? this still requires detailed explanation.
Statistical analysis: I suggest to insert a measure of the magnitude of the effect for the comparisons. Please consider to include effect sizes.
Ethical Issue: please declare the approval number and the name of the ethical competent body.
Discussion: I also suggest improving the discussion, it must be updated with the comparison and discussion regarding the post pandemic context of physical activity in the general population, a paragraph should be added with a proper reference. Also, the section of limitations and future search is also very short, the Authors could elaborate on that.
Author Response
Thank you for giving us the opportunity to submit a revised draft of our manuscript titled “Perceptions, attitudes, and practices of Greek Athletes towards COVID-19 vaccination and its consequences on sports training and sports activity” /with proposed new title “ Evaluation of the perceptions, attitudes, and practices among Greek non-professional athletes visiting a public hospital during March 2022, towards COVID-19 vaccination and its consequences on sports training and sports activity”, to Vaccines Special Issue "Epidemiology, Vaccination and Public Health". We appreciate the time and effort that you and the reviewers have dedicated to providing your valuable feedback on our manuscript.
Point 1: Title: It should be improved, highlighting the main object of the study, period and population.
Response 1: We thank the Reviewer for this comment. To best accommodate your comment, we have accordingly modified the Title.
Point 2: Introduction: The authors should make it clear about what is the gap in the literature that is filled with this study. The study must be actualized by framing it within the vast body of literature that addressed the COVID-19 related changed habits in the context of the main 21st-century challenges (refer to articles with DOI: 10.3390/ijerph191911929).
Response 2: We thank the Reviewer for his/her kind remark. We have accordingly modified the Introduction section. REVISED: (INTRODUCTION, p.2, lines 61-62;67-69)
Point 3: Methods: This section raises the main scientific concerns. The questionnaire is a non-standard tools. The use of an unreliable instrument is a serious and irreversible limitation. The fact that a similar methods have been used in previous studies is not sufficient. A validation process must be performed and reported to evaluate the tool. What about intelligibility, reliability and validation index?
Response 3: We thank the Reviewer for this comment. Questions about COVID-19 were created by the authors and based on previous questionnaires. The questionnaire was forward translated from English to Greek by three independent translators. Greek was the mother tongue of these translators. The prefinal version of the translated questionnaire was tested on a small group of 10 people from the target population. After completing the questionnaire, the respondents were interviewed to ensure that the questionnaire was understandable and presented accurately the meaning of the original questions. To accommodate the comment of the Reviewer, this information is now provided in detail in the Methods section and the limitations are discussed in the relevant section of the Discussion. REVISED (MATERIALS AND METHODS, p.3, lines: 122-132; DISCUSSION, p.11, lines 378-383)
Point 4: The authors do not propose a minimum sample size, what is the reference population? The Authors talk about patients of an hospital, what is the study population?
Response 4: Firstly, it is important to clarify that Laiko General Hospital is one of the biggest hospitals in the region of Greece having a specialized sports medicine doctor; thus, a great number of people involved with sports are visiting the hospital. Aiming to best address your comment, additional information on the reference population have been included in our Methods. REVISED (MATERIALS AND METHODS, p.3, lines 115-118)
Point 5: Without the numerical identification of the reference population is not clear the validity of the study. A non-representative sample is by its self a non-sense-survey. How did they avoid selection bias? How was questionnaire distributed? It is not clear how participants were recruited? Did thy Authors pay some company to provide lists of subscribers or distribute the investigation tool? this still requires detailed explanation.
Response 5: To accommodate these comments as much as possible, we have now added in the Methods section a detailed description of the procedure that was followed. REVISED (MATERIALS AND METHODS, p.3, lines 104-110; lines 115-119)
Point 6: Statistical analysis: I suggest to insert a measure of the magnitude of the effect for the comparisons. Please consider to include effect sizes.
Response 6: Sex-specific odds ratios (OR) and 95% confidence intervals (95%CI) are reported to depict the magnitude of the associations. The paragraph has been revised accordingly. REVISED (Materials and Methods, p. 4, lines 162-163; Results, p. 8, lines 245-253)
Point 7: Issue: please declare the approval number and the name of the ethical competent body.
Response 7: As suggested by the reviewer, we have accordingly modified the section. REVISED (Institutional Review Board Statement, p.11, lines: 415-417)
Point 8: Discussion: I also suggest improving the discussion, it must be updated with the comparison and discussion regarding the post pandemic context of physical activity in the general population, a paragraph should be added with a proper reference. Also, the section of limitations and future search is also very short, the Authors could elaborate on that.
Response 8: To best accommodate your comment, we have accordingly modified Discussion and limitations paragraph. REVISED (DISCUSSION, p.10, lines 339-347; p.11, lines 370-374, lines 375-388)
We look forward to hearing from you in due time regarding our submission and to respond to any further questions and comments you may have. We thank the reviewer for his/her thoughtful and thorough review and believe his/her input has been invaluable to make our manuscript more balanced.
Sincerely,
The corresponding author on behalf of all authors
Round 2
Reviewer 2 Report
The paper was improved and it is now suitable for pubblication.